# Hepatitis C Virus Translation Regulation

**DOI:** 10.3390/ijms21072328

**Published:** 2020-03-27

**Authors:** Michael Niepmann, Gesche K. Gerresheim

**Affiliations:** Institute of Biochemistry, Medical Faculty, Justus-Liebig-University, Friedrichstrasse 24, 35392 Giessen, Germany; Gesche.Gerresheim@gmx.de

**Keywords:** HCV, *Hepacivirus*, internal ribosome entry site, IRES, initiation, ribosome, 40S, eIF3, ITAF, stress response

## Abstract

Translation of the hepatitis C virus (HCV) RNA genome is regulated by the internal ribosome entry site (IRES), located in the 5’-untranslated region (5′UTR) and part of the core protein coding sequence, and by the 3′UTR. The 5′UTR has some highly conserved structural regions, while others can assume different conformations. The IRES can bind to the ribosomal 40S subunit with high affinity without any other factors. Nevertheless, IRES activity is modulated by additional *cis* sequences in the viral genome, including the 3′UTR and the *cis*-acting replication element (CRE). Canonical translation initiation factors (eIFs) are involved in HCV translation initiation, including eIF3, eIF2, eIF1A, eIF5, and eIF5B. Alternatively, under stress conditions and limited eIF2-Met-tRNA_i_^Met^ availability, alternative initiation factors such as eIF2D, eIF2A, and eIF5B can substitute for eIF2 to allow HCV translation even when cellular mRNA translation is downregulated. In addition, several IRES trans-acting factors (ITAFs) modulate IRES activity by building large networks of RNA-protein and protein–protein interactions, also connecting 5′- and 3′-ends of the viral RNA. Moreover, some ITAFs can act as RNA chaperones that help to position the viral AUG start codon in the ribosomal 40S subunit entry channel. Finally, the liver-specific microRNA-122 (miR-122) stimulates HCV IRES-dependent translation, most likely by stabilizing a certain structure of the IRES that is required for initiation.

## 1. Introduction

Hepatitis C virus (HCV) is an enveloped positive strand RNA virus that preferentially replicates in the liver [1], and it is classified in the genus *Hepacivirus* in the family *Flaviviridae*. Worldwide, about 71 million people are infected with HCV [2]. The infection is usually noticed only when coincidentally diagnosed by routine testing, for example, during hospitalization, or when the liver disease becomes acute. In the latter case, liver damage by virus replication and the resulting immune responses can lead to impaired bilirubin conjugation in the liver, and unconjugated bilirubin deposits can then be noticed as a yellowish color (called jaundice), often first in the sclera in the eyes and when more severe also in the skin. An acute infection can result in severe liver damage, in rare cases even resulting in death [3,4]. However, most HCV infections remain inapparent [5,6], and the virus infection can become chronic in about 60% to 70 % of all infections [7], often without being noticed. Chronic infection can, in the long run, result in liver cirrhosis and liver cancer (hepatocellular carcinoma, HCC) [8,9,10], while a metabolic reprogramming of the infected cells according to the “Warburg effect” like in cancer cells can be observed only a few days after the onset of HCV replication [11]. Moreover, inapparent replication of the virus usually results in unnoticed spread of the virus to other individuals, a fact that is a major challenge for surveillance, health care, and treatment [12]. Meanwhile, very effective treatment regimens using direct acting antivirals (DAAs) are available, although they are still very expensive [13,14]. Although the error rate of the viral replicase is high and can, in principle, easily give rise to resistance mutations, the conserved nature of the replicase active center and the often occurring reduced fitness of mutants result in the rare appearance of resistance mutations against nucleoside inhibitors such as sofosbuvir [15,16]. An effective vaccination is not yet available, also partially due to the high variability of the viral RNA genome. Thus, further research on HCV is urgently required to combat HCV infections, and despite much progress in the understanding of HCV replication the molecular mechanisms of HCV replication are still far from being completely understood [12].

HCV molecular biology research essentially started with the first cloning of the HCV genome (“non-A non-B hepatitis”) [17] (Figure 1). Further important landmarks were the development of a subgenomic replicon system [18] that first analyzed intracellular virus replication, and the development of infectious full-length clones that were capable of going through a complete viral replication cycle including infection and new virus production [19,20,21].

The HCV RNA genome is about 9600 nucleotides (nts) long and codes for one long polyprotein that is co- and post-translationally processed into the mature gene products [29,30,31]. The viral structural proteins include the core protein, which is contained in the virus particle, as well as the envelope glycoproteins E1 and E2. The non-structural (NS) proteins are involved in HCV RNA replication and particle assembly. The first NS protein is p7, a viroporin, which is involved in assembly. NS2 is a protease and acts as a cofactor involved in assembly. NS3 has protease and helicase functions, and the protease activity of NS3 is responsible for the cleavage of downstream NS protein precursor cleavage sites. NS4A is a NS3 cofactor, whereas NS4B is involved in the membrane reorganization of replication complex formation. NS5A is involved in RNA replication and assembly, and NS5B actually is the viral replicase (RNA-dependent RNA polymerase, RdRp) [29,30,31].

The HCV virus particle comes as a lipo-viro-particle [32,33]. After binding to a variety of surface receptors including the low-density lipoprotein (LDL) receptor [33,34,35] and entry into the cytoplasm, the viral NS proteins produced in the first (“pilot”) round of translation recruit cellular membranes that originate from the endoplasmic reticulum (ER) and form a so-called membranous web in which the viral proteins and genomes form spatially coordinated replication complexes [31,36,37]. Each incoming plus strand which survived cellular immune responses [5,38,39] and degradation is replicated by the NS5B replicase and gives rise to one antigenome minus strand copy, which in turn generates about 10 progeny plus strands [30]. HCV RNA synthesis is regulated by a variety of RNA signals that reside close to the very 3′- and 5′-ends of the genome, but also sequence and RNA secondary structure elements in the coding region (largely in the 3′-terminal NS5B coding sequence) contribute to the regulation of RNA synthesis [23,30,40,41,42,43,44,45]. Bulk translation from the progeny plus strand genomes, then, yields a vast excess of viral proteins over the number of genomes [30,46]. Then, progeny plus strand RNA genomes together with viral proteins are packaged into newly assembled virions [32,47], together with some cellular proteins such as apolipoproteins (Apo) A-I, B, C-II, and E, which contribute to liver tropism of the virus by binding to the next infected hepatocyte´s surface receptors [48].

Unlike most cellular mRNAs, the 5′-end of the HCV genomic RNA has no cap nucleotide attached which would govern efficient cap-dependent translation initiation [49,50,51]. Instead, HCV translation is mediated by virtue of an internal ribosome entry site (IRES) [22,52,53,54,55], which is located largely in the 5′UTR but also slightly spans into the coding region (Figure 1). While the resulting low efficient translation coincides with the “undercover” strategy of HCV replication that often leads to chronic infection and further unnoticed spread of the virus to uninfected individuals, the use of such IRES elements has two more big advantages. The first advantage is that in particular the very ends of the RNA genome do not need to serve functions in translation control such as in capped and polyadenylated cellular mRNAs. Instead, RNA signals that are involved in genome replication can be directly placed at the very genome ends [23,30,45]. The second benefit of cap-independent translation is that the virus escapes antiviral countermeasures of the cell in terms of the downregulation of cap-dependent translation, which is largely conferred by phosphorylation of eIF2 and the resulting inhibition of cap-dependent translation initiation [56].

In addition to the hepatocyte surface receptors mentioned above, another determinant of the liver tropism of HCV is the microRNA-122 (miR-122). microRNAs are small single-stranded guide RNAs that direct effector complexes involving Argonaute (Ago) proteins to cellular mRNAs and usually negatively influence the mRNA´s translation efficiency and induce degradation of the mRNA [57,58]. miR-122 is expressed almost exclusively in the liver and constitutes about 70% of all microRNAs in hepatocytes [59], whereas it is nearly not expressed in other tissues [60]. In contrast to the negative influence of microRNAs on cellular mRNAs, HCV utilizes miR-122 to promote its own replication [61], thereby making the liver-specific miR-122 another determinant for the liver tropism of HCV. There are five to six target sequences conserved in the HCV genome, two in the 5′UTR [62], one in the 3′UTR, and two to three in the NS5B coding region (depending on genotype) [63]. Cooperative binding of two miR-122 molecules to the two adjacent target sites in the HCV 5′UTR contributes to RNA stability by protecting against cellular nucleases [64,65] and has a positive effect on the efficiency of HCV translation [66,67,68,69,70,71,72]. Although some studies investigated the possible roles of the conserved potential miR-122 binding sites in the NS5B coding region and in the 3′UTR, it is not yet clear if physical binding of miR-122 to these sites results in effector functions [73,74,75,76], leaving some doubts why these sequences are conserved among HCV isolates.

In this review, we focus on the sequences, cellular factors, and molecular mechanisms involved in the regulation of translation by the HCV IRES. Thereby, we touch the functions of miR-122 specifically only with regard to translation regulation, while another review by Joyce Wilson in this review series thoroughly covers miR-122 action in all aspects of HCV replication.

## 2. An Overview over HCV Genome Regions Involved in Translation Regulation

The RNA *cis*-elements that control HCV RNA genome translation largely reside in the 5′ and 3′ regions of the genome (Figure 1). The IRES element is located in the 5′UTR plus some 30 nts of the core coding region (dotted line in Figure 1) [52,53,77,78]. It contains the large branched domain (Dom) III (or stem-loop (SL) III, respectively) with several small subdomain stem-loops, the SL IV and a double pseudoknot, and the upstream SL II. In contrast, the SL I located at the very 5-end of the genome is not involved in translation regulation but in replication control. The sequence between the SLs I and II, which contains the two miR-122 binding sites located directly upstream of the IRES, is formally not counted as belonging to the IRES, but this sequence can influence translation activity upon binding of miR-122 [66]. The IRES binds the small ribosomal 40S subunit with high affinity of about 2 nM [79] (underlayed in yellow in Figure 1).

At the 3-end of the HCV genome, the 3′UTR [80], as well as the *cis*-acting replication element (CRE) with its stem-loops 5BSL3.2 and 5BSL3.3 in the 3′-terminal region of the NS5B coding region [81,82] are also involved in translation regulation (Figure 1). The function of the 3′UTR and the possible function of the CRE in translation regulation are related to two types of long-range interactions (LRIs), first RNA–RNA LRIs, and second those long-range interactions that are mediated by RNA-binding proteins or by the ribosomal 40S subunit. This 3′- to 5′-end communication is reminiscent of that occurring with cellular mRNAs. There, the polyA tract at the 3′-end stimulates translation at the 5′-end. In principle, this interaction in *cis* can only indicate that the RNA to be translated is not degraded but intact, and only then it makes sense to translate this RNA efficiently.

Reports about a contribution of the HCV 3′UTR to the regulation of translation at the IRES have been quite different in the past. Some previous reports have shown a positive influence of the 3′UTR on IRES-directed translation [80,83,84,85,86,87]. One report showed a negative influence of the 3′UTR [88], and other studies reported no influence [89,90,91,92]. These discrepancies could have been caused by the use of different reporter systems and artificial extensions at the reporter RNA´s 3′-end (discussed in [86]). Currently, it is rather clear that the 3′UTR actually stimulates translation, and it is recognized that suitable reporter assays should use RNA transfection (not DNA) and a precise authentic 3′-end of the HCV 3′UTR [86].

The CRE (see Figure 1, also called SL5B3.2) was first described to be involved in control of overall HCV genome replication [93] and binds physically to the NS5B replicase [94,95,96]. Moreover, the CRE 5BSL3.2 plus the flanking downstream 5BSL3.3 [82], as well as the poly(U/C)-tract of the 3′UTR [82,97] were shown to bind to the small 40S ribosomal subunit. Thereby, the CRE binds with K_D_ of about 9 nM to the 40S subunit, while the poly(U/C) tract of the 3′UTR binds even stronger (K_D_ = 1 nM); together, the CRE and 3′UTR bind 40S subunits with a comparable affinity as the IRES [82,97].

Long-range RNA–RNA interactions are very important in HCV replication. Several such putative interactions have been described [24,45,63,82,98,99,100,101,102,103,104,105,106,107,108]. By far the most important interactions (see Figure 1) that have been demonstrated to be functionally relevant by several studies are the following. First, the interaction between the apical loop of the CRE 5BSL3.2 and the apical loop of the SL 2 in the 3′UTR (also named “kissing loop” interaction) [98,99] is important for HCV replication. Second, the interaction between the bulge of the CRE 5BSL3.2 (GCCCG) with a sequence about 200 nts upstream (CGGGC) (“9170” in Figure 1 and in [23]) was also shown to be important for replication [100,103] and third, the internal CRE 5BSL3.2 bulge can alternatively interact with the apical loop of the SL IIId (UGGGU) in the IRES [101].

Regarding translation regulation, conflicting results have been reported with respect to a possible role of the CRE. One study showed that deletion of the CRE 5BSL3.2 conferred an increase of translation efficiency from HCV-luciferase translation reporter constructs 18 h after transfection (but not after 6 h) [102], suggesting that the CRE 5BSL3.2 is involved in inhibiting HCV translation and, together with the flanking upstream 5BSL3.1 and the downstream 5BSL3.3, the CRE 5BSL3.2 inhibits translation [102]. This implicates that this LRI could have something to do with translation regulation or with a possible switch between translation and replication. In contrast, another study showed that the inhibition of the CRE by hybridization with locked nucleic acid (LNA) oligonucleotides impairs translation of luciferase reporter RNAs or replication-incompetent replicons 6 h after transfection [81], leading to the conclusion that the CRE rather stimulates translation. This discrepancy could be due to differences in the reporter assay systems and yet needs to be further elucidated.

In addition, a hybridization between sequences in the core-coding region and the region between 5′UTR SLs I and II (red in Figure 1) had been described, which has a negative effect on translation efficiency. This interaction is important in terms of IRES structure, function, and miR-122 action and is discussed in the next section.

## 3. Structure of the HCV 5′UTR and the Internal Ribosome Entry Site

The HCV 5′UTR is about 340 nts long and is predicted to fold into characteristic RNA secondary structures (Figure 2). These sequences and the predicted secondary structures are highly conserved among HCV genotypes and subtypes [63]. 

The small SL I form very close to the very 5′-end of the HCV 5′UTR, leaving only three or four unpaired nts at the very 5′-end. In the canonical representation of the 5′UTR secondary structure (Figure 2A) [42,63,109,111], this SL I is followed by a single-stranded stretch of about 42 nts which binds two molecules of miR-122 [62] with one of the miR-122 molecules also hybridizing to the very 5′ nts upstream of SL I. This entire sequence often is named domain I of the 5′UTR. Downstream of that, the canonical representation depicts the SL II (or domain II), followed by a large branched domain that is called either SL III or domain III, which contains the subdomains or SLs IIIa, b, c, d, e, and f. This domain III, then, is followed by a single-stranded stretch that can form a pseudoknot 1 (PK1) with the SL IIIf [116]. In addition, a U residue (postion 300 in Figure 2) the SL IIIe can form one base pair with the upstream A (position 291), creating a second pseudoknot (PK2). This region with the two pseudoknots forms a compact and unique tertiary structure [117]. The downstream SL IV contain the AUG start codon [118]. These IRES RNA secondary structures have been predicted in silico and have been experimentally validated by chemical structure and nuclease protection mapping [52,63,77,113]. Moreover, the ribosome-bound HCV IRES structure has been validated by cryo electron microscopy (EM) [55,119,120,121]. An alternative predicted 5′UTR structure shows a refolded, alternative SL II (SL II^alt^) (Figure 2B), which is as consistent with the reported experimental structure mapping results as is the canonical structure [72,77,113,115]. In this alternative fold of the 5′UTR sequences between SL I and SL III (Figure 2B), the two miR-122 binding sites are largely hidden within double-stranded RNA structures, and therefore are less accessible as compared with the canonical structure (Figure 2A). In the presence of miR-122, the alternative SL II^alt^ reforms to adopt the canonical structure, allowing efficient translation and stabilization of the genome [72,115,122]. Interestingly, also mutations that render HCV replication miR-122 independent favor the canonical structure [72,122].

Despite the seemingly clearly defined 5′UTR structure, we can distinguish some central structure-driving regions from others, which are more flexible and can change structure, also according to functional requirements. We have performed our own in silico predictions using LocARNA [123] and RNAalifold [124] with a representative number of different isolates from different genotypes (selected from [63]) (data not shown). On the one hand, some IRES regions appear to have a quite strong preference to robustly fold into a definite structure, likely driven by the intrinsic properties of the conserved primary sequence. Formation of these RNA secondary structures takes place regardless of subtle variations in their sequence that lead to covariations in their secondary structure, and regardless of the availability of varying flanking RNA regions that could interfere with the secondary structures by providing options for alternative interactions. On the other hand, other sequences appear to be able to dynamically assume different secondary structures, depending on the extent of flanking RNA that is available for interaction, depending on the binding of miR-122, and depending on the binding of cellular proteins as well as of the ribosomal 40S subunit.

In the first group of sequence elements, i.e., those which appear to robustly form a defined and invariant secondary structure, we would list essentially three regions (drawn in bold in the 5′UTR structures in Figure 3). In brief, these are the SLI, the upper part of domain III, and the base of the domain III with the double pseudoknot. In part, the structure of these regions can also be stabilized by binding to cellular factors or to the ribosomal 40S subunit. The first region is the G/C-rich stem-loop I near the very 5′-end of the HCV 5′UTR. The SL I region has a strong preference to form under almost any conditions such as varying length of flanking sequences or subtle sequence variations among genotypes and subtypes, with only very few exceptions. The second sequence region that appears to always robustly form the same structure in in silico predictions is the upper part of the domain III, including the four-way junction with the stem-loops IIIa, IIIb, and IIIc [125]. This structure binds eIF3 [79] and contains conserved elements in its central bulge (around position 180 and 220 in Figure 2A) as well as in the stem above which are required for eIF3 binding [126,127,128]. The third steadily forming region includes the base of the domain III with SL IIIe, IIIf, and the double pseudoknots [116,117], which can drive binding to and be stabilized by the ribosomal 40S subunit [129]. These three robustly forming 5′UTR regions can also influence the biased folding of flanking sequences, and small-angle X-ray scattering of the HCV IRES in solution, and in silico structure flexibility simulations [114] are consistent with an overall IRES structure as predicted. In addition, the first three RNA secondary structures in the core coding region also appear to have a conserved tendency to robustly fold in silico to form the SLs V, VI, and the following SL 588. The formation of these distinct structures are required to leave only the SL IV sequence to form either a stem-loop or to unfold and bind in the ribosome entry channel, without being disturbed by intruding flanking sequences (Figure 3E).

In contrast, other sequences are more flexible. This is particularly true for the region between SLs I and III. In the canonical 5′UTR structure (Figure 3A), the region between SLs I and II with the two miR-122 binding sites is predicted to be single stranded in RNAalifold structure predictions, and the canonical SL II structure forms. This structure is predicted to occur by default also in the absence of miR-122 (i.e., in the absence of constraints to keep the miR-122 target sites as single-stranded sites in the in silico foldings). Thus, the region between SL I and SL II is available to easily bind miR-122. In turn, miR-122 binding stabilizes this region in single-stranded form (Figure 3A), then supporting the stable formation of SL II and its important interactions with the apical loop of SL IV and with the ribosomal 40S subunit [72,115]. These structure changes of SL II induced (or stabilized, respectively) by miR-122 can account for the stimulation of translation by miR-122 [66]. However, an only very slight variation of folding parameters (e.g., by not running RNAalifold but instead running LocARNA with default parameters) changes the prediction dramatically, with the sequences downstream of SL I largely rearranging and partially covering the miR-122 target sites. Then, the alternative SL II^alt^ structure forms (Figure 2B and Figure 3B), and the miR-122 target sites are largely hidden. This indicates that this region is very flexible in structure and can easily refold in response to the absence or presence of miR-122, thereby changing the structure and by that the function of the SL II sequence region which is involved in regulating translation.

In addition, a sequence in the core coding region (nts 428 to 442) can fold back to the miR-122 binding region in a long-range interaction and cover the miR-122 target sites, forming a more compact structure (Figure 3C, and also see the red LRI in Figure 1) and reducing translation efficiency [130,131,132,133]. This inhibitory hybridization can be relieved by miR-122 binding [135,136], while the physiological relevance for controlling HCV translation in cells appears not to be dramatic but can more easily be detected in an in vitro translation system [134].

A yet completely hypothetical structure is the structure shown in Figure 3D. In this structure, the lower part of the domain III is refolded largely to loop the SL IIId and form the alternative SL IIId* [63]. This prediction was yielded as the most stable consensus structure among the 106 HCV isolates from all available genotypes and subtypes [63]. However, its minimum fold energy (MFE) is only marginally lower than that of the canonical form. Since all RNA structure probing experiments and cryo EM structures, as mentioned above, are not consistent with 5′UTR structure D but rather with structure A, we can only speculate if this conserved structure D could have possible functions.

The predicted secondary structure of the SL IV with the AUG start codon contains covariations among isolates, which implies that this small secondary structure must be of some importance. However, the position of the start codon within the predicted SL IV secondary obviously suggests that there must be changes in RNA secondary structure when the start codon is positioned in the ribosome. When the IRES binds the 40S subunit, the SL IV unfolds and is placed in the entry channel of the 40S subunit (Figure 3E) [117,137,138], preparing the IRES-40S complex for translation initiation.

With respect to the previously described interchangeable 5′UTR/IRES structures and their possible biological relevance (Figure 3), as well as with regard to the required detachment of the IRES RNA from the ribosome upon completion of the translation initiation cycle, we need to take into account that under natural conditions in the cell the secondary and tertiary structure of the IRES RNA could be much more flexible than suggested by the seemingly rigid, fixed canonical structure, as shown in Figure 2A and Figure 3, structure A. Many IRES structural studies (in the absence or presence of the 40S subunit) have been performed at quite high magnesium concentrations. Moreover, binding of the 40S subunit to the IRES can strongly favor a distinct IRES structure by induced fit. Most studies used 2.5 mM MgCl_2_ [55,113,125,139,140,141,142,143], whereas others used 3 mM [97], 4.5 mM [119], 5 mM [114,138], or even 10 mM MgCl_2_ [144]. In contrast, the concentration of free Mg^2+^ in the cytosol in different cells is 0.86 mM on average over many studies, and in rat liver cells, the free intracellular Mg^2+^ concentration is approximately 0.7 mM (reviewed in [145]).

The high Mg^2+^ concentrations used in the studies mentioned above have demonstrated the IRES and ribosome structures previously reported (see above). However, at such high Mg^2+^ concentrations, IRES binding to the 40S subunit is strongly favored [79], leaving no options for an equilibrium of different IRES structures in solution. Moreover, in the presence of 60S subunits, both ribosomal subunits would be largely associated at 2.5 mM Mg^2+^ [146], and by that they would even not allow binding of the HCV IRES to individual 40S subunits, followed by the induction of 60S subunit association by the HCV RNA to be translated. In contrast, at the physiological concentration of 0.7 mM free Mg^2+^, only about half of the IRES molecules are actually completely folded [79], while the other half assume more flexible structure intermediates. In addition, another study showed that the IRES has a rather extended conformation at low Mg^2+^ concentrations, whereas the pseudoknots form only in the presence of Mg^2+^ [136]. Concurrently, HCV translation is optimal at Mg^2+^ concentrations lower than 1 mM [147]. Thus, we must be aware that the functional HCV IRES structure is much more flexible in terms of opening and closing secondary structures than most studies on IRES structure suggest.

The above considerations particularly apply to those IRES regions that must dynamically interact with the 40S subunit (drawn thin in the IRES structures in Figure 3). Although the HCV IRES is completely unfolded in the absence of Mg^2+^, strongly folding IRES core structures such as the upper portion of SL III (shown in bold in Figure 3) form properly into the canonically shown form at Mg^2+^ concentration of only 0.25 mM and above [110,136] (i.e., are formed under intracellular conditions). In contrast, the G residue 135 which is the second nucleotide in the left part of the base of domain III (a region that is routinely drawn as double-stranded in the IRES secondary structure predictions [63]) is largely protected (i.e., double-stranded) at Mg^2+^ concentrations of 2.5 mM, whereas it appears only 50% protected at the physiological Mg^2+^ concentration of 0.7 mM [110]. These results imply that the HCV IRES structure can be (and perhaps must be) much more flexible during productive complete translation initiation cycles than many studies suggest. This gives room to the speculation that the previously mentioned highly conserved predicted structure D (Figure 3) could have some biological relevance yet to be shown. One of the most intriguing questions for future research is how the 40S subunit, which initially binds very strongly to the IRES manages to routinely detach from the IRES in order to commence the transition from translation initiation to elongation and synthesis of the polyprotein. This likely needs to be investigated at magnesium concentrations that are lower than those used in previous studies (best 0.7 mM).

## 4. Contacts of the HCV IRES with the Small Ribosomal 40S Subunit and with eIF3

When the protein-free HCV IRES is allowed to bind in vitro to isolated small ribosomal 40S subunits, the IRES makes several close contacts to the 40S subunit. We must assume that under conditions of intracellular magnesium concentrations (as discussed above) there should be more flexibility of the IRES [110]. However, we are not aware of studies that analyzed the sequential arrival of different IRES regions at the 40S subunit with high time resolution (in the sub-second scale), and thereby demonstrated structural “induced fit” changes of the IRES structure during binding. The only study that methodically comes close [148] shows that there is a long lag phase until the IRES binds the 40S subunit.

Nevertheless, from the IRES structure as it appears when bound to the isolated 40S subunit (Figure 4) [55,119,121,138,149], we can assume that there are three distinct regions of the IRES that are different in binding, as well as in function. The first region is the “core” of the IRES, including the two pseudoknots PK1 and PK2 together with the base of domain III and SLs IIId, IIIe, and IIIf, and considered in an extended version also including the four-way junction SL IIIabc which also binds closely to the 40S subunit (but not including the actual SL IIIb). This “core” IRES region essentially serves to “anchor” the body of the IRES with high affinity in a fixed position (irrespective of possible induced fit changes during binding) on the 40S subunit and provides a platform for the flexible connection of the other two functional IRES modules, SL II and SL IIIb.

In contrast, the second functional IRES region, the SL II region in its canonical form (see Figure 2 and Figure 3A,E), appears to be rather flexible and fulfills important tasks in reorganizing the IRES structure and the ribosome in order to unwind the SL IV, place the contained AUG start codon in the ribosomal mRNA entry channel, and manipulate the 40S subunit to undergo initiation (see below).

The third region is the SL IIIb. It appears not to bind to the 40S subunit at all [55,119,121,138,149], but it is connected to the rest of the IRES by the very flexible SL IIIabc four-way junction [151] which is anchored on ribosomal protein eS27 [138]. This SL IIIb is used by the IRES solely to bind eIF3 [126,127] after the IRES has displaced eIF3 from its binding to the 40S subunit [143]. The reasons for this are not yet fully understood.

However, it appears likely that the IRES needs to keep eIF3 on hold in close vicinity of *cis* for using its contacts to the HCV 3′UTR [97], to avoid premature subunit joining [152,153], and to use it later in subsequent initiation steps to acquire eIF2, eIF5B, and the 60 subunit to the initiation complex [141,154,155]. In addition, the AUG start codon of the HCV IRES must be properly positioned in the 40S mRNA entry channel in order to fully accommodate its contacts on the 40S subunit for proper initiation [137]. However, it has been reported that eIF3 is not absolutely essential for subsequent stages in initiation, since 48S complexes formed in the absence of eIF3 on both wild-type and SL IIIb mutant HCV IRES elements readily underwent subunit joining, forming elongation-competent 80S ribosomes [143]. Nevertheless, the HCV IRES can still bind eIF3 also in the complete 80S ribosomes [156], likely in the same way with the IRES SL IIIb.

The IRES core domain with the double pseudoknot, SL IIId, IIIe, and IIIf tightly contacts the 40S subunit on the solvent side [117,129] by binding to the ribosomal 18S rRNA, as well as to several ribosomal proteins. The HCV IRES SL IIId apical loop GGG sequence contacts a CCC sequence in the 18S rRNA helix 26 ES (expansion segment) 7 apical loop (close to the 40S subunit mRNA exit tunnel) [119,138,143,150,157], while also SL IIIe makes a contact to this helix expansion segment [138,150]. Several specific ribosomal proteins are also known to be contacted by the IRES core domain and SL III, listed in the following (for the new nomenclature of ribosomal proteins see [158], where the prefix “u” stands for ribosomal proteins universal to bacterial, archeal, and eukaryotic ribosomes, and “e” stands for ribosomal proteins unique to eukaryotes). The ribosomal protein eS27 contacts the upper IRES four-way junction with SLs IIIa and IIIc and the SL IIId [138,143,144,159]. Proteins eS1, uS7, uS9, and uS11 contact the lower part of the SL III with SLs IIId and IIIe [138,159,160], and proteins eS10, eS26, and eS28 (near the ribosomal exit channel) contact the IRES more downstream at the double pseudoknot and at the beginning of the core coding region [138,143,159,161]. In contrast, the IRES SL II contacts ribosomal proteins uS7, uS9, uS11, and uS25 [119,138,144,160,161,162].

The IRES SL II functions in preparing the ribosomal 40S subunit for initiation. Whereas IRES domain III and pseudoknot contacts to the IRES appear to function primarily in tight binding, the domain II of the IRES exerts important functions in manipulating the ribosome and to facilitate the positioning of the AUG start codon region in the 40S entry channel and subsequent 60S subunit joining [121,141,150]. Therefore, the SL II needs to be flexible, a feature that is mainly conferred by included bulges to allow flexible changes [112,163]. The apical part of SL II is essential for ribosomal subunit joining [154]. This part contacts the 40S subunit in the region of the head and the edge of the platform, near the mRNA entry channel [55,138,149], and causes a slight rotation of the 40S head and changes in the structure of the platform [55]. Thereby, SL II occupies similar binding sites as the E-site tRNA and eIF2 [138]. SL II helps to unfold the SL IV with the AUG start codon and to position it in the mRNA entry channel in the 40S-bound conformation [137,149], perhaps also by base-pairing with the region with the start codon like tRNA [121,138] (see Figure 2 and Figure 3). Finally, the SL II stimulates eIF5-mediated hydrolysis of eIF2-bound GTP and joining of a 60S subunit [141,164,165]. Consistently, a deletion of three nucleotides (GCC) from the apical loop of SL II completely abolishes SL II function on the 40S subunit and results in an accumulation of 80S ribosomes after 15 min [162], suggesting that assembled 80S ribosomes are arrested on such defective IRES and are unable to undergo the transition from initiation to elongation, which normally takes place after about 6 min [66].

Interestingly, the HCV IRES is able to bind to translating ribosomes (which translate regular cap-dependent cellular mRNAs). Thereby, the IRES binds with its “core” described above (SL IIIacdef plus PK1 and PK2), but without SL IIIb and SL II, to the solvent side of the 40S subunit in the translating 80S ribosome. In this way, the IRES “hitchhikes” with an actively translating ribosome until regular termination of the cap-dependent mRNA occurs. After termination, the HCV IRES is already present on the 40S subunit in *cis* and efficiently usurps the post-termination 40S subunit [166].

## 5. Steps Involved in HCV Translation Initiation

When we consider the order of binding events taking place during translation initiation at the HCV IRES RNA (Figure 5), it is important to note that the affinity of the HCV IRES to the isolated small ribosomal 40S subunit is much higher than its affinity to isolated eIF3. The HCV IRES can bind to the small ribosomal 40S subunit independently of any initiation factors [167]. The dissociation constant (K_D_) of IRES binding to isolated 40S subunits is about 2 nM, whereas the K_D_ of IRES binding to isolated eIF3 is only about 35 nM [79,141,154]. The SL II contributes only very little to the overall IRES-40S affinity [79,141]. The high affinity of the IRES-40S interaction is caused by multiple and very close contacts of several IRES regions with the 40S subunit [55,119,121,138,149], whereas binding to eIF3 includes only the apical region of the SL III, in particular the SL IIIb [79,126,127,141,143].

These affinities could suggest that the HCV IRES first binds to naked 40S subunits, and only after that, the IRES can additionally acquire eIF3 (as shown in Figure 5A, upper left arrows). This idea emerges from several studies that analyzed the binding of the HCV IRES to purified 40S subunits [55,79,119,121,141,149,154,169]. However, eIF3 is known to bind to 40S subunits obtained from post-termination complexes, and then remains routinely bound to the 40S subunits in order to facilitate the next initiation round [170,171,172]. Thus, we need to consider that most 40S subunits are available as 40S-eIF3 complexes. Moreover, eIF3 wraps around nearly the entire 40S subunit, including the 60 subunit interface [173], and largely covers exactly those regions on the surface of the 40S subunit that are supposed to also bind the HCV IRES, or the very similar IRES of CSFV (classical swine fever virus) [143]. In this respect, it should be noted that in [120], the binding positions of HCV IRES and eIF3 were just artificially overlayed in silico and shown in the same figure, leading to the possible misunderstanding that HCV IRES and eIF3 could simultaneously bind to essentially the same position on the 40S subunit.

However, the CSFV IRES (which functionally acts in the same way as the HCV IRES) appears to displace eIF3 from its binding position on the 40S subunit. Thereby, the IRES effectively usurps ribosomal contacts of eIF3 [174]. Then, large parts of the IRES bind closely to the 40S subunit, and eIF3 is only indirectly kept bound in the complex solely by contacting the IRES RNA, but not any more by contacting the 40S subunit [143] (compare Figure 4B with A). Surprisingly, binding of the HCV IRES to the preformed 40S-eIF3 complex is essentially not impaired by the presence of eIF3 on the 40S ribosomes, but the presence of eIF3 appears to facilitate IRES binding to the 40S subunit [79]. Thus, although the position of eIF3 on the 40S subunit could be considered to sterically hinder IRES binding, eIF3 somehow facilitates IRES binding instead of competing with it. Taken together, we can assume that under the in vivo conditions in the cell, the natural substrate for binding of the HCV IRES is the 40S-eIF3 complex (Figure 5A, upper right arrow).

eIF2 is the standard factor that routinely delivers the charged initiator tRNA (Met-tRNA_i_^Met^) to the initiation complex that binds to AUG start codons of most cellular mRNAs [175,176,177,178]. The addition of eIF2 largely facilitates formation of ribosomal initiation complexes with the CSFV IRES [167]. The charged initiator Met-tRNA_i_^Met^ is also required for efficient complex formation with the HCV IRES [167]. Binding of eIF2-Met-tRNA_i_^Met^ to preinitiation complexes is facilitated by the eIF3 subunit eIF3a [155]. Efficient 48S initiation complex formation also requires eIF1A, whereas eIF1 interferes with 48S initiation complex formation [165,179]. Domain II of HCV-like IRESs stimulates eIF5-mediated hydrolysis of eIF2-bound GTP and joining of the 60S subunit [141,164,165]. eIF5 serves to remove discharged eIF2 from the initiation complex (Figure 5A, lower part). Subsequent formation of 80S complexes with 60S subunit joining additionally requires eIF5B [165], which is recruited to the IRES-40S complex by the eIF3 subunit eIF3c [155].

From a kinetic point of view, the association of the HCV IRES with ribosomes is a rather slow process. For comparison, with the highly efficiently translated cap-dependent β-globin mRNA, formation of complete 80S ribosomes was detected to be maximal after 15 s (the first time point that was analyzed in that study), and even the second, third, and fourth wave of 80S ribosomes had already been loaded to the mRNA after 15 s [180], giving rise to corresponding polysomes at that early time point. In contrast, the association of the HCV IRES with the 40S subunit is rather slow. Low amounts of the resulting 48S initiation complexes could be detected after 1 min [66,154], but the formation of maximal amounts of the first wave of 48S initiation complexes required 3 to 6 min [66,154]. Thereby, the association of some molecules of the HCV IRES RNA with some 40S subunits can occur within seconds, but saturating binding of most molecules of IRES RNA and 40S subunits in the populations takes about 40 to 80 s [148]. The presence of miR-122 can greatly accelerate and enhance this process [66]. Formation of the first wave of complete 80S ribosomes requires 4 to 6 min [66,154]. Between 6 and 10 min, the first wave of 80S complexes leaves the initiation site, and the second wave of 48S complexes forms [66]. These kinetic differences, despite high affinity binding to the 40S subunit, could contribute to the relatively low translation efficiency of the HCV IRES as compared with cellular mRNAs [181].

## 6. Use of Alternative Initiation Factors under Stress Conditions

eIF2 is one of the main targets of general translation regulation in the cell. Such regulation takes place at the initiation step of translation and occurs during different stress conditions such as starvation, ER stress, or after activation of innate immune responses during a viral infection. Under such conditions, the α-subunit of eIF2 is phosphorylated by eIF2 kinases, and eIF2 tightly associates with eIF2B, resulting in inactivation of eIF2 [56]. As a consequence, translation of most cap-dependent cellular mRNAs is downregulated [49,50,51,175].

However, under such conditions the HCV RNA can still be translated with sufficient efficiency to allow viral protein synthesis [182]. When eIF2-GTP-Met-tRNA_i_^Met^ ternary complex availability is reduced, translation initiation at the HCV IRES switches to eIF2-independent modes of translation initiation (Figure 5B,C), and in some cases also independent of the initiator tRNA_i_^Met^ [178,183]. Then, translation initiation at the HCV IRES is mediated by alternative initiation factors. Several protein factors have been proposed for this role including eIF2A [184], eIF2D [183], eIF5B [185], a combination of eIF2A and eIF5B [186], and the complex of the proteins MCT-1 and DENR [187].

The 139 kDa eIF5B (the eukaryotic homolog of bacterial IF2) [188,189] can promote formation of 80S complexes with the HCV IRES initiator-tRNA binding to the ribosomal P site in the presence of only additional eIF3 [51,149,185] (Figure 5B). Thereby, eIF5B substitutes for eIF2 and eIF5, and 80S complexes formed without eIF2 are competent for translational elongation [185]. In addition, for the closely related CSFV IRES, translation initiation can occur using the same mechanism [165].

eIF2A (a protein of about 65 kDa) was described in 1975 and was characterized as a factor that is capable of GTP-independent binding of Met-tRNA_i_^Met^ to the 40S subunit of eukaryotic ribosome [190]. Meanwhile, there are conflicting reports about the possible role of eIF2A. Cloned eIF2A (NCBI nucleotide database entry NM_032025) has been described to have Met-tRNA binding properties and is also able to deliver Met-tRNA_i_^Met^ in HCV translation initiation [184,186]. This eIF2A interacts with the HCV IRES core domain including SLs IIId, IIIe, IIIf, and the pseudoknots, with specific binding determinants present in the SL IIId, and eIF2A relocates from the nucleus to the cytoplasm in HCV-infected cells [184], suggesting its importance to function as an eIF2 surrogate during HCV infection under stress conditions. In contrast, in another study it was discovered that the activity formerly attributed to eIF2A was performed by another protein that copurified along with the previously described eIF2A over almost the entire purification procedure, while after final separation steps the purified eIF2A did not show any such activity. The new protein was, then, named eIF2D (NM_006893) (formerly named “ligatin” by mistake) and facilitated the delivery of Met-tRNA_i_^Met^ to the P-site of the 40S subunit independent of GTP (see below) [183]. The same eIF2D was described to facilitate HCV translation initiation [187]. According to a recent study [186], eIF2A can also function synergistically with eIF5B. Both eIF2A and eIF5B can bind to the 40S subunit during stress conditions [186]. eIF5B interacts both with eIF2A and with the tRNA, and eIF5B augments the activity of eIF2A in loading Met-tRNA_i_^Met^ onto a 40S ribosome associated with an HCV [186].

Alternatively, eIF2D or a set of two other related proteins, MCT-1 and DENR, can facilitate translation initiation at the HCV IRES (Figure 5C). eIF2D has also a molecular mass of about 65 kDa, a fact that led to the previously mentioned confusion with eIF2A when the activity of eIF2D at the HCV IRES was first described [183]. It should be noted that, unlike eIF2A, eIF2D resembles other initiation factors since it contains a domain similar to translation initiation factor eIF1 [183]. Recent work performed with ribosomal profiling indicates that eIF2D is involved in post-termination events, where it promotes ribosome recycling [191]. eIF2D can confer HCV translation initiation as an alternative to eIF2. eIF2D but not eIF2 can utilize non-AUG codons in the HCV IRES [183]. On other RNAs, eIF2D can also bind to non-AUG codons in the P-site [178], suggesting that in the case of non-AUG ARF translation (see below), eIF2D can substitute for eIF2. Interestingly, eIF2D does not need to be loaded with initiator Met-tRNA_i_^Met^ or even with any tRNA to bind to the ribosomal P-site [183] and causes 40S and 60S subunit joining in the absence of eIF5B [187,192]. The interaction of eIF1 and eIF1A with the 40S subunit interferes with the binding of eIF2D to the 40S subunit, whereas eIF3 binding does not [187]. As an alternative to eIF2D, the heterodimeric complex of MCT-1 (the product of malignant T cell-amplified sequence 1 oncogene) and DENR (density regulated protein) [192] can substitute for eIF2D in delivering the charged tRNA_i_^Met^ to the ribosome [168,187,192].

However, a very recent study reported that the HCV IRES still works efficiently under conditions of a suppressed eIF2 activity in double knockout cells lacking both eIF2A and eIF2D [191]. If so, this result is more consistent with the option to use only eIF5B for HCV translation initiation under stress conditions (Figure 5B, left). It also makes the diagrams in Figure 5B (middle and right) and Figure 5C involving alternative factors eIF2A, eIF2D and, possibly, DENR and MCT-1, somewhat less attractive.

Anyway, HCV translation is able to escape the suppression of general cellular translation under conditions of stress or viral infection when eIF2 activity is downregulated by phosphorylation. Nevertheless, it is worthwhile to note that even under conditions using eIF2 for initiation, the efficiency of translation directed by the HCV IRES is much lower than that of average cap-dependent cellular mRNAs [181]. This leaves HCV translation at a level that does not completely perturb cellular gene expression and does not flood the cell with viral products, and thus allows for ongoing long-term low-level HCV replication during chronic infection.

## 7. IRES Trans-Acting Factors (ITAFs)

Several cellular proteins, which are not routinely involved in translation initiation of cellular mRNAs, are recruited by the HCV RNA and modulate its activity, either in translation regulation or in replication. Some of these proteins are also involved in regulating the activity of picornavirus IRES elements [54,193]. Here, we focus on those cellular proteins that are involved in translation regulation. These proteins (Figure 6) have been called IRES trans-acting factors (ITAFs), a term which is also used here, but this term should not be regarded too strictly, since some of these factors also interact with the HCV 3′UTR or with other RNA genome regions.

Most of these proteins bind directly to the RNA, whereas a few others stimulate HCV translation indirectly, or we do not yet know how exactly they modulate IRES activity. The roles of many of these proteins have been reviewed in detail before (please see [54,193]). Most of these proteins have multiple RNA-binding domains (shown in Figure 6) and form homo- and heterodimers, or they even act as multidomain protein complex organizers such as Gemin5 [196]. Moreover, some of the proteins bind not only to the 5′UTR/IRES but also to the 3′UTR. By that, these proteins can be assumed to build a large network of RNA–protein and protein–protein interactions that connects the HCV RNA genome 5′- and 3′-ends, supported by direct binding of the 40S subunit and eIF3 to both the 5′- and 3′-regions (see above, and Figure 7). Proteins involved in this network which directly bind to the HCV RNA are La [197], NSAP1 [198], hnRNP L [199] and D [200], IMP1 [156], PCBP2 [201], the Lsm1-7 complex, and the negatively acting Gemin5 [196,202], and perhaps also PTB [203,204] and RBM24 [205]. Interestingly, the above proteins bind to many sites on the HCV plus strand RNA, but the very 3′end of the genomic RNA is not covered, suggesting that the 3′end is left available for the initiation of RNA minus strand synthesis by the NS5B replicase, likely supported by the NFAR proteins [206]. In contrast, the NFAR protein complex (NF90, NF45, and RHA) appears not to be involved in HCV translation regulation [207], even though it binds to both the HCV 5′, and 3′UTRs and is involved in replication [208]. Network components which do not directly bind to the HCV RNA but participate by protein–protein interactions are HuR (ELAVL1) [209,210], the proteasome subunit α7 (PSMA7) [209,211], and perhaps PatL1, a P-body component involved in mRNA degradation that decreases HCV translation reporter gene expression [212].

In addition to building a large interaction network, a second possible function of these ITAFs could be an RNA chaperone function. For example, for a picornavirus IRES we have shown that PTB, using its different RNA-binding domains, connects different parts of the large IRES structure, and thereby stimulates IRES activity [213]. Such a function appears not so likely for the HCV IRES, which has a rather compact structure. However, we could speculate that some of these ITAFs support the unfolding of the HCV IRES SL IV and by that facilitate the entrance of this sequence in the 40S subunit mRNA entry channel. Specifically, La protein and IMP1 bind to the HCV sequence region involving the PK1 and the SL IV [156,197,214,215], and La protein can probably support melting of the SL IV [216]. Perhaps, these proteins induce a single-stranded conformation of this IRES region, and thereby help to position the AUG start codon in the mRNA entry channel of the 40S subunit. In addition, NSAP1 [198,217] and hnRNP L [199,218] bind the HCV RNA downstream of the AUG and the SL IV (Figure 7, and also refer to Figure 4). Thereby, NSAP1 was reported to promote the correct positioning of the 40S ribosomal subunit at the initiation codon [217]. Thus, we can speculate that NSAP1 and hnRNP L bind to the HCV RNA downstream of the 40S entry channel similar to pulling hands on a rope to keep the unfolded single-stranded RNA in the 40S entry channel, and by that block snap-back folding of the SL IV which could then slip out of the entry channel.

The third category of ITAFs involved in HCV translation can alter RNA structure. The representative of this category is the RNA helicase DDX6 (also called RCK or p54). There are conflicting reports regarding the role of DDX6 in HCV overall replication. Two reports showed that DDX6 has a positive role in HCV translation [69,212], and one of these reports claimed that the positive effect of DDX6 on HCV translation was independent of miR-122 [69]. In contrast, another report claimed that DDX6 has no influence on HCV translation, while HCV RNA stability depended on DDX6 (mediated by DDX6-dependent binding of miR-122 to the second 5′UTR binding site) [219]. This is consistent with a previous study that claimed a positive effect of DDX6 on HCV replication but not translation [220].

The fourth category of proteins has only indirect regulatory influence on HCV translation, while it is assumed that these proteins are no components of protein complexes directly acting on HCV RNA. Identified by a siRNA screen using a HCV IRES reporter RNA, MAP kinase interacting serine/threonine kinase 1 (MKNK1) and phosphatidylinositol 4-kinase catalytical subunit beta (PI4K-beta) were identified to stimulate HCV translation [221], consistent with a previous report that had described a general positive effect of PI4K-beta on overall HCV replication [222].

## 8. Expression of the Alternative Reading Frame ARF/core+1

For about two decades, a number of reports have claimed that, in addition to the canonical polyprotein ORF, another protein is produced from the core ORF region in the core+1 frame, the “alternative reading frame” (ARF) or “core+1” (originally also called “F”) protein (see Figure 1 and Figure 8). The perception of the possible importance of this protein has been hampered for long time by the following three circumstances: (1) The reading frame for this putative protein is only moderately conserved among HCV genotypes and isolates and has a variety of start and stop codons at different positions; (2) the mechanism of initiation of its translation was difficult to elucidate, and its expression is relatively weak; and (3) the possible function of these putative proteins appeared enigmatic for long time, while only recently some progress has been achieved in this direction.

The genetic structure of the ARF/core+1 ORF appears to be very variable among HCV genotypes and isolates (Figure 8). The start codons for initiation of these proteins were identified to be essentially at codons 26, 85, and 87 of the core+1 reading frame, when the nucleotide No. 5 (AUGAG) of the core ORF is counted as the first nucleotide of the core+1 ORF [223,224,225,226,227,228,229], but also codon 58 was reported to give rise to another version of the ARF protein [228]. We have analyzed the ARF/core+1 ORF sequences of each two isolates of HCV genotypes 1, 2, 3, 4, and 6 (10 isolates in total, selected from the MAFFT alignment in [63]), including the often used genotype isolates 1b “Con1”, 2a “J6”, and 2a “JFH” sequences (Figure 2B). The result is that the ARF/core+1 ORF is open from codon 26 to at least codon 124 of the ARF/core+1 frame, i.e., for at least 99 amino acids, confirming previous results obtained with many isolates among virtually all HCV genotypes [25,26,27,223,228,230]. From codon 125, stop codons are interspersed, except in genotype 1a which has a stop only after codon 161. The start point at codon 26 gives rise to the core+1/L protein [229], whereas the start at codon 85 yields the core+1/S protein [223], thus, producing proteins with two different conserved N-termini but heterogeneous C-termini due to genotype- or isolate-specific stop codons.

Moreover, the ARF/core+1 ORF appears much less conserved than the overlapping canonical core ORF. Analyzing the above-mentioned 10 selected isolates, we find that the codons 26 to 124 of the core+1 frame (i.e., the core+1/L protein) have 99 AAs, of which 27 AAs (27.3%) are identical at their positions and an additional 20 AAs (20.2%) are similar, resulting in an overall similarity of 47.5%. The codons 26 to 161 of the core+1 frame (i.e., the core+1/L protein) have 136 AAs, 34 AAs (25%) identical, plus 25 (18.4%) similar, resulting in 43.4% overall similarity. The codons 85 to 161 of the core+1 (i.e., the core+1/S protein) frame have 77 AAs, 21 AAs (27.3%) identical, plus 10 (13%) similar, resulting in 40.3% overall similarity. Thereby, the conservation of the ARF/core+1 protein is somewhat stronger near its N-terminus and directly at its shortest consensus C-terminus (compare Figure 8B). However, the ARF/core+1 frame is by far less conserved than the regular core ORF (71.7% identical, plus 18.8% similar AAs, overall similarity 90.5%).

Regarding the mechanism by which translation of the ARF/core+1 frame is conferred, initially some evidence was presented for ribosomal frameshifting to produce the ARF/core+1 protein, named “F” [26]. The authors fused an HA tag to the N-terminus of the canonical core ORF and found a shift in the core+1 ORF, indicating expression of the ARF. The efficiency of expression was very high in rabbit reticulocyte lysate (RRL) but was found to be only about 1% in Huh-7 hepatoma cells. The authors speculated that an A/C-rich sequence around core+1 of codons 8 to 14 could be responsible for this frameshifting [26]. However, this comparison of translation systems shows that the RRL translation machinery (derived from reticulocytes that develop to erythrocytes, a cell system with very reduced protein complexity) is quite flexible in using RNA templates, while more complex cells exhibit more stringent translation control. The activity in RRL can now be regarded as collateral background expression by unspecific RNA binding by the translation machinery [232].

In contrast, introduction of stop codons between the putative frameshifting site around codons 8 to 14 and codon 85 did not abrogate ARF protein production in cells, strongly arguing against frameshifting, but for alternative internal use of downstream codons such as codon 85 or 87 [223,225,226,227,228]. In addition to core+1, codon 85, codon 26 can also be used for initiation [224,229]. Moreover, reduction of canonical core protein ORF expression increased ARF expression, whereas an increase of canonical core frame expression reduced ARF expression, again arguing against the frameshift hypothesis [223,225,233]. The core+1 start codons 85 and 87 often have AUG (but also ACG or ACC), whereas the codons 26 and 58 only have non-AUG codons (GUG, GCG, and GAG), whereas all codons No. 26, 58, 85 and 87 are located in a moderately strong Kozak context [234].

The possible function of the ARF/core+1 protein was unclear for long time, and only recently some functions were described. Short-term in vivo experiments in Huh-7 cells, in SCID mice carrying primary human hepatocytes and even in chimpanzees did not reveal evidence for a functional role of a core+1 frame product in JFH1 (HCV genotype 2a) virus production [235,236]. However, antibodies against the core+1 product could be detected in patient sera ([25,27,237] and references in [230]). During acute HCV infection, seroconversion to anti-core+1 antibodies can be observed [238]. Patients with chronic HCV infection, liver cirrhosis, and hepatocellular carcinoma have many more antibodies against core+1 products than other patients, and the antibody titers correlate with the extent of liver cirrhosis [239,240,241]. These findings show that ARF/core+1 actually is expressed in chronic HCV infection of the liver, and the extent of ARF/core+1 expression (likely as overall HCV expression) in the liver correlates with the extent of liver damage.

Some rather mechanistic studies shed some light on how the HCV ARF/core+1 protein can support HCV replication and by that confer a long-term advantage for the virus. However, the emerging picture appears not yet fully consistent. An early study showed that p53 and p21 promoters were activated when ARF protein was overexpressed [28], a finding that would argue for ARF proteins acting as tumor suppressors. In contrast, activation of the hepcidin promoter by the transcription factor AP-1 was inhibited by the ARF/core+1 protein [242], which is supposed to result in reduced inhibition of iron export from enterocytes, leading to iron overload and increased oxidative stress in the body [243], but the possible advantage for HCV yet needs to be shown.

Two more described functions indicate how expression of the ARF/core+1 protein could provide evident advantages for HCV replication in the body. Overexpression of the ARF/core+1 protein suppresses the expression of interferon-stimulated genes (ISGs), including the pattern recognition receptor retinoic-acid-inducible gene-I (RIG-I) [244]. This indicates that the ARF/core+1 protein contributes to establishing long-term HCV replication in the liver. Another clear evidence for a role of the ARF/core+1 protein comes from the analysis of cell cycle control. Overexpression of the HCV genotype 1a (H) protein in two isoforms resulted in slightly higher expression of cyclin D1 and in strongly enhanced phosphorylation of the retinoblastoma (Rb) protein, correlating with higher cell proliferation rates in Huh-7.5 cells [245]. Concomitantly, expression of cellular proto-oncogenes including hras, c-fos, c-jun, c-myc, and vav1 was elevated under ARF/core+1 overexpression, and the number of tumors in mice overexpressing the ARF/core+1 protein during chemically induced tumorigenesis was significantly increased [245]. This indicates that HCV not only rapidly reprograms the hepatocyte metabolism to promote the Warburg effect that is a characteristic of tumor cells [11,246] but also establishes proto-oncogene expression changes that lead to cancer in the long run. Future research should also systematically search for cellular binding partners of the ARF/core+1 protein by global scale interactome and for targets of gene expression regulation by crosslink immunoprecipitation (CLIP) studies. Even though non-AUG start codons can be used with high efficiency under normal conditions [247], it could also be interesting to find out if the largely non-canonical initiation codons used for ARF/core+1 expression could also be used by eIF2A and eIF2D which are known to support non-canonical initiation under stress conditions [178,183].

Taken together, after decades of research, more and more facets of possible ARF/core+1 protein functions in the infected body appear, but the view of its functions is far from complete, a situation that is reminiscent of another small “accessory” protein in another virus infecting the liver, hepatitis B virus (HBV) [248].

## 9. Future Directions

In the past, we have learned a lot about the function of IRES and the factors involved in that. In part, this has been mostly based on technical progress in Cryo EM visualization techniques. On the basis of this, as well as on classical molecular biological and biochemical methods, it would be interesting to further investigate IRES structure and the transition from initiation to elongation, analyzed at intracellular magnesium concentrations. This includes the visualization of the IRES structure mediated by miR-122/Ago complexes and the actual implications for SL II action, as well as the visualization of ITAFs (e.g., NSAP1, hnRNP L, La, and hnRNP D) associated with the IRES on the 40S subunit in order to elucidate their function. Further questions for future directions could aim at the functional implications of the “eIF3 holding” function of the ribosome-bound IRES, as well as the further elucidation of the diversity, structure(s), and functions of the ARF proteins, in particular the molecular details of the regulation of gene expression and the long-term implications for HCC.

## Figures and Tables

**Figure 1 ijms-21-02328-f001:**
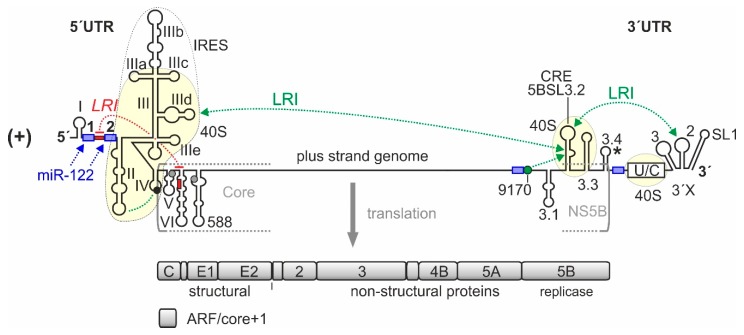
*cis*-Elements in the hepatitis C virus (HCV) RNA genome that are involved in translation regulation. The HCV plus strand RNA genome. The internal ribosome entry site (IRES) in the HCV 5′-untranslated region (5′UTR), the entire 3′UTR and the *cis*-acting replication element (CRE) in the NS5B coding region are involved in translation regulation [22,23,24]. Those regions of the 5′UTR, 3′UTR, and CRE that bind to the ribosomal 40S subunit are underlayed in light yellow. Stem-loops (SLs) in the 5′UTR are numbered by roman numerals. The region of the IRES is surrounded by a dotted line. The IRES includes SLs II–IV of the 5′UTR but spans into the core protein coding region. The canonical AUG start codon in SL IV of the 5′UTR (black circle) gives rise to translation of the polyprotein which is cleaved to yield structural proteins and non-structural (NS-) proteins, including the viral replicase NS5B. The 3′UTR contains the variable region, a poly(U/C) tract (U/C), and the so-called 3′X region including SLs 1, 2, and 3. The stem-loop 5BSL3.2 in the 3′-region of the NS5B coding region is the CRE, flanked by upstream stem-loop 5BSL3.1 and downstream 5BSL3.3. The polyprotein stop codon is located in the stem-loop 5BSL3.4 (asterisk). Some other start codons which give rise to the alternative reading frame (ARF) in the core+1 reading frame [25,26,27,28] are shown in grey. Positively and negatively acting long-range RNA–RNA interactions (LRIs) are shown in green or red, respectively, with the sequence “9170” shown as green circle. Selected binding sites for microRNA-122 (miR-122) are shown in blue.

**Figure 2 ijms-21-02328-f002:**
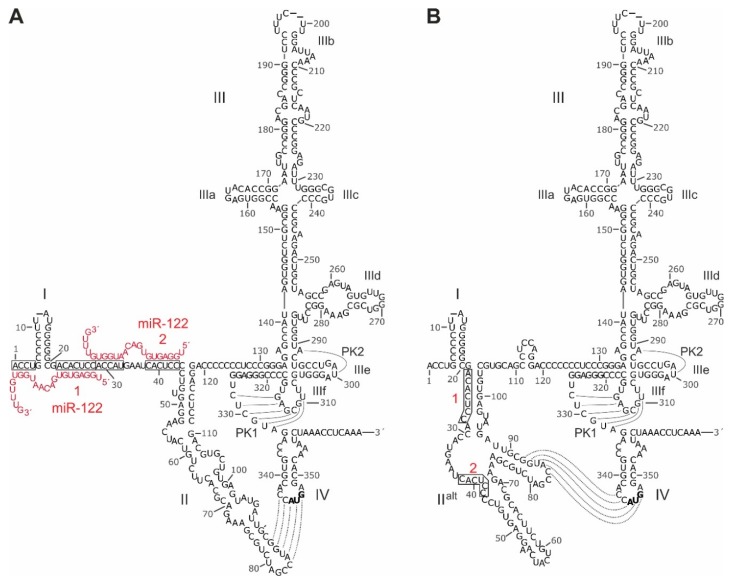
Sequence and RNA secondary structure of the HCV IRES. (**A**) The 5′UTR of HCV and its canonical structure [63,109,110,111,112,113,114]. The sequence shown is from genotype 2a (J6 isolate), with length variations among HCV isolates indicated by short horizontal dashes at nucleotide positions. Nucleotide numbering is according to the MAFFT alignment in the supplement of [63] (http://www.rna.uni-jena.de/supplements/hcv/). For comparison, the first nucleotide of the core coding sequence (AUG, in bold) in the SL IV is nucleotide No. 342 in genotype 1b (Con1) and No. 341 in genotype 2a (J6 and JFH1 isolates). IRES SL domains are indicated by roman numerals. microRNA-122 (miR-122, red) binding is indicated, miR-122 target sequences are boxed. Pseudoknot (PK) base pairing and the interaction between SL II and SL IV is indicated; (**B**) Secondary structure model of the 5′UTR with alternative folding of SL II^alt^. The structure is largely according to [72,77,115], with minor modifications according to our RNAalifold outputs using several genotypes (not shown).

**Figure 3 ijms-21-02328-f003:**
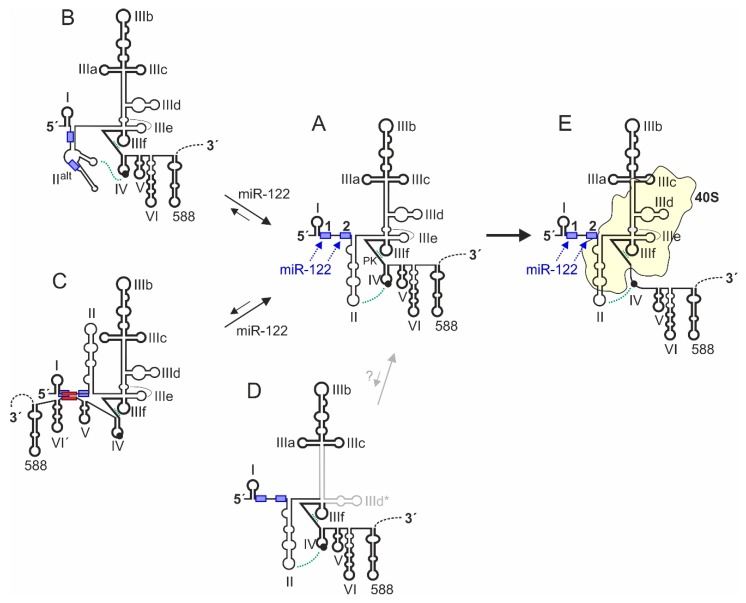
Conformational dynamics of HCV 5′UTR structure. The conserved canonical 5′UTR structure [63,109,110,111,112,113,114] is shown in the middle (structure A). 5′UTR sequences which have a strong intrinsic tendency to fold into a distinct secondary and tertiary structure are depicted in bold. The constitutive double pseudoknot (PK) interactions involving SL IIIe (PK2) and IIIf (PK1) and the possible interaction between SL II and SL IV are indicated. The 5′UTR structure B [72,77,115] has the region between SLs I and III refolded and forms an alternative version of SL II (SL II^alt^). This SL II^alt^ can form in the absence of miR-122, then hiding both miR-122 binding sites. Binding of miR-122 promotes refolding of structure B to structure A [72,115,122]. The 5′UTR structure C can form by fold-back of nucleotides 428–442 in the base of SL VI in the core protein coding region to hybridize with sequences downstream of SL I. In this state, the translation activity of the IRES is decreased [130,131,132,133,134]. This hybridization also interferes with binding of miR-122 to the 5′UTR. In turn, binding of miR-122 induces refolding of structure C to structure A [135]. Structure D is a yet completely hypothetic structure that was predicted to be the energetically slightly favored conserved structure [63]. In this structure D, the SL IIId of the IRES is refolded to form SL IIId*. However, a possible biological relevance of this hypothetical structure is unclear, in particular since several interactions of the canonical SL IIId have been described (see main text). Structure E is formed upon binding to the ribosomal 40S subunit. Then, SL IV sequences containing the AUG start codon are unfolded and positioned in the mRNA entry channel of the 40S subunit, aided by interaction with SL II.

**Figure 4 ijms-21-02328-f004:**
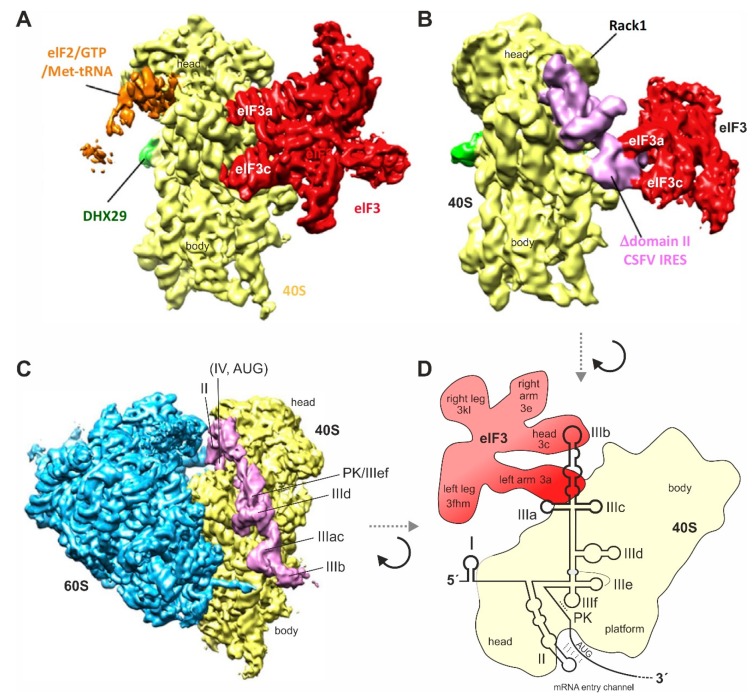
Binding of the HCV IRES to the ribosome and to eIF3. (**A**) The small ribosomal 40S subunit (yellow) with bound eIF3 (red). eIF3 makes multiple contacts to the 40S subunit using several subunits, including eIF3a and eIF3c; (**B**) The IRES of classical swine fever virus (CSFV) (pink) without IRES domain II, binding to the 40S subunit (yellow) and to eIF3 (red). The IRES has displaced eIF3 completely from its binding to the 40S subunit (compare panel A) but keeps it connected to the 40S subunit only indirectly by contacts between IRES SL IIIabc and eIF3. Figures A and B were reprinted from [150] (Figure 2F) and slightly modified. Reprinted with permission from Elsevier (licence no. 4761840166773); (**C**) The HCV IRES (pink) binding to the 40S subunit (yellow) in the complete 80S ribosome (60S subunit in blue). The IRES SL IIIdef/PK is in close contact with the 40S subunit, the SLs II and IV are positioned in the mRNA entry channel, and the SL IIIb is pointing to the solvent side for binding eIF3. The IRES is not touching the 60S subunit [121,138]. Figure C was modified from the left panel of Figure 1C in [121]. Reproduced with permission from EMBO; (**D**) Schematic illustration of the HCV IRES binding to the 40S subunit (yellow) and to eIF3 (red) when eIF3 is largely displaced from the 40S subunit by the IRES, similar as in (B). The orientation of the 40S subunit is shown essentially top-down as compared with (A–C), viewed approximately from the solvent side.

**Figure 5 ijms-21-02328-f005:**
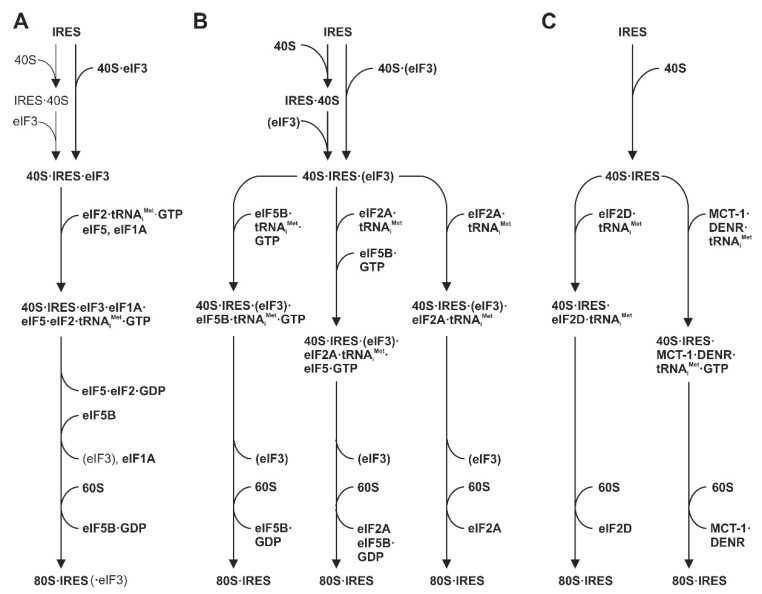
The steps involved in translation initiation at the HCV IRES. (**A**) The canonical way of translation initiation with the HCV IRES under conditions of sufficient eIF2 activity. In vivo, the IRES most likely binds to 40S-eIF3 post-termination complexes (top right), while in vitro-studies also suggest that binding of 40S and eIF3 can occur subsequently (top left). Then, the ternary 40S-IRES-eIF3 complex acquires eIF2 charged with tRNA_i_^Met^ and GTP, as well as eIF1A and eIF5. After locating the HCV AUG start codon, eIF5 catalyzes release of decharged eIF2-GDP from the ribosome. eIF5B then causes subunit joining, and eIF5B-GDP, eIF1A, and eIF3 leave the complex which is, then, ready for the first translation elongation step. “Hitchhiking” of the IRES on translating 80S ribosomes is not shown (see main text); (**B** and **C**) Alternative translation initiation pathways for the HCV IRES under stress conditions leading to eIF2α phosphorylation, i.e., under limited eIF2 availability. (B) Binary IRES-40S complexes, which can or cannot also bind eIF3, bind either eIF2A (left), eIF5B (right) or both eIF2A and eIF5B in combination (middle). In the cases when eIF2A is present, it delivers the tRNA_i_^Met^. In the absence of eIF2A, eIF5B delivers the initiator tRNA. (C) The function of tRNA_i_^Met^ delivery can also be taken over by either eIF2D or by two proteins which together are structured similar to eIF2D, namely MCT-1 and DENR [168].

**Figure 6 ijms-21-02328-f006:**
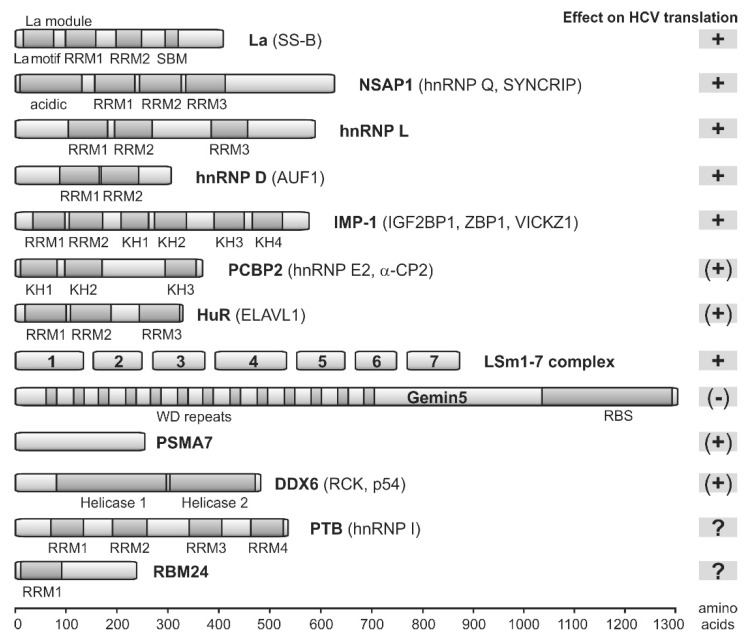
IRES trans-acting factors (ITAFs) that modulate HCV IRES activity. Important functional domains are shown in dark grey. In most cases, these domains are RNA-binding domains, similar to the RNA-recognition motif domain (RRM) [194] or the K-homology (KH) domains of hnRNP proteins [195].

**Figure 7 ijms-21-02328-f007:**
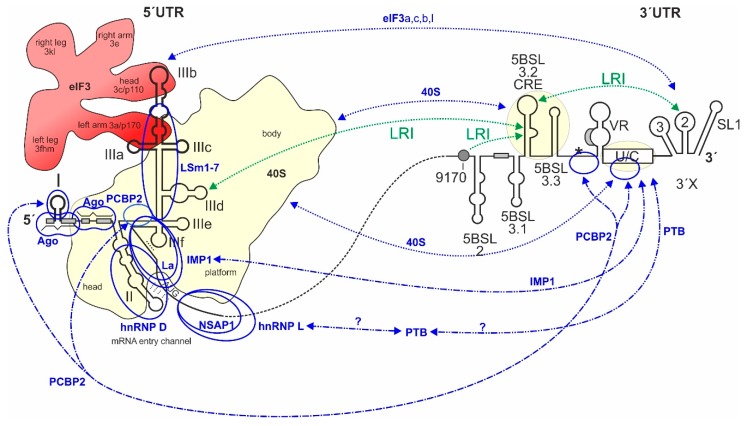
Interactions between HCV 5′- and 3′UTR. The HCV RNA is shown as a black line, with most of the coding region shortened as a dotted line. RNA–RNA long-range interactions (LRIs) are shown with green lines. Interactions mediated by proteins, by the 40S subunit, and by eIF3 are shown with blue lines. The proteins are shown as transparent bodies with blue lines. Other indications are as in previous figures. For details, please see main text.

**Figure 8 ijms-21-02328-f008:**
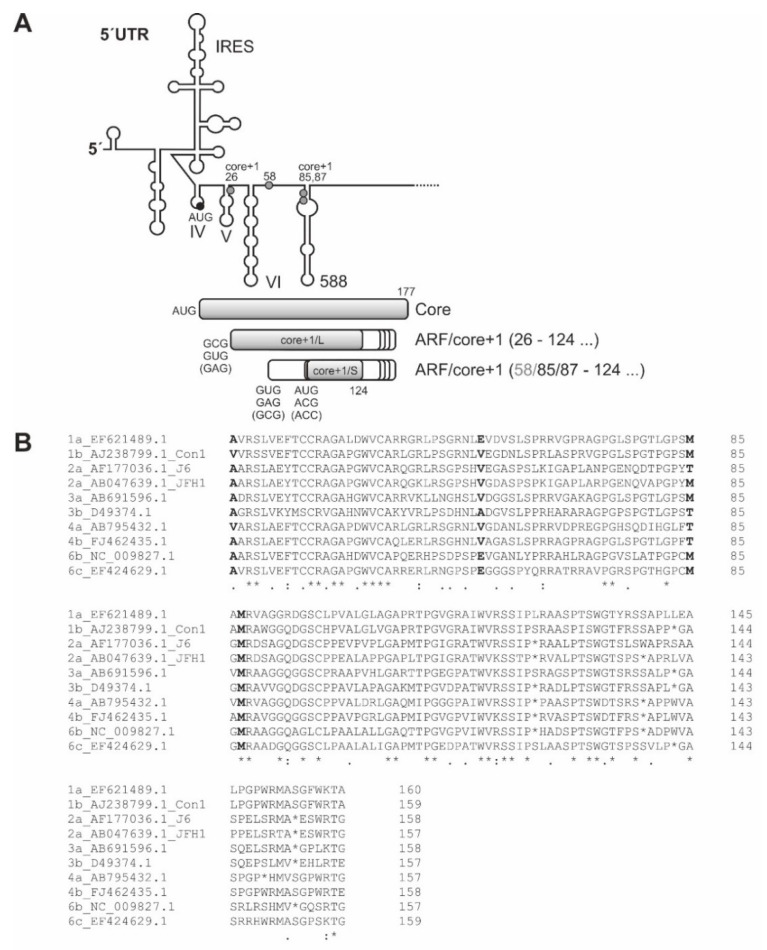
Translation of the HCV alternative reading frame (ARF) or core+1 frame. (**A**) Overview over starts and stops that result in expression of a variety of different ARF of core+1 protein, among HCV isolates. The HCV 5′UTR and the downstream core region is shown essentially as in Figure 1. The start AUG for the canonical core protein is shown in SL IV as black filled circle. The start codons of core+1 protein are shown as grey filled circles at the core+1 frame for positions 26, 58, 85, and 87. The canonical core protein and the various core+1 ARF products are shown as boxes below. The major products produced in genotype 2a are shown in grey, other products produced by initiation at codon 58 or by termination at other more downstream stops in other genotypes are underlayed in white. Start codon usage is depicted; (**B**) Amino acid sequences of representative HCV genotypes and subtypes (selected from [63]); genotypes and NCBI nucleotide database accession numbers are given on the left, as well as abbreviations of some well-known isolates. The AA sequence starts with codon 26 of the core+1 frame, when the nucleotide No. 5 of the canonical core frame, AUGAG, is nucleotide No. 1 of the core+1 frame. The amino acids encoded by the main start codons 26, 58, 85, and 87 are in bold, stop codons are shown by asterisks in the sequence. Conservation is shown under the alignment, with (*) indicating absolute conservation, (:) indicating strongly similar and (.) indicating weakly similar AA properties. The dot indicating similarity between AAs at position 7 in terms of being charged but neglecting charge reversal have been removed, since charge reversal can have serious consequences for proteins [231].

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
