# Peer review of "Hepatitis C Virus Translation Regulation"

_ijms, 2020, doi:10.3390/ijms21072328_

Round 1

Reviewer 1 Report

This review provides an in-depth description of how Hepatitis C virus translation is regulated by several regions of the HCV genome and the history of their identification. It also describes alternative structures formed by the 5’ UTR and IRES and how they might be influence by annealing of miR-122, and also provides novel predictions and interpretations of potential 5’ UTR RNA structures and new models of how miR-122 might modulate them. It then provides an overview of how the HCV IRES initiates translation under ‘normal’ conditions and how its regulation is modified during cellular stress or antiviral responses that down regulate translation. It also describes the canonical translation factors involved and how IRES trans‐acting factors (ITEFs) might regulate the HCV IRES and mediate interactions between the HCV genomic 5’ and 3’ untranslated regions. The review is extremely thorough and generally well written. I suggest only a few minor revisions to the content and English as listed below.

Minor issues:

Line 34: the person infected with ʺJapanese Fulminant Hepatitisʺ isolate 1 did not die. I believe they were treated and cured in spite of the high HCV replication levels displayed during that specific infection. This sentence should be modified based on the information in reference 3.

Line 44. ‘high error rate of the viral replicase can easily give rise to resistance mutations’. While this is true the low fitness of some resistant viruses has limited the impact of resistance thus far during patient treatment. Resistance might be an issue in the future, but it does not seem to be a huge issue now. It is very different than HIV treatment resistance. This sentence might be tempered to reflect this.

Line 55, ‘cap’ nucleotide should be capped nucleotide

Line 190 ‘In addition, a negative hybridization between sequences in the core‐coding region and the 191 region between 5´UTR SLs I and II (red in Figure 1) had been described’ what is a NEGATIVE hybridization? Please clarify.

Line 224. Other reports predicted and analyzed the 5’ UTR structure and predicted miR-122’s influence on the IRES structure and function and should be referenced. PMID 30053137, PMID 30941417. In addition, the alternative structure vs the canonical structure of SLI and SLII has not been confirmed experimentally. In papers that analyzed the RNA structure using SHAPE the results were consistent with both the canonical and the alt structures.

The figure reference should be a consistent format. In places the sections of Figure 3 are referenced with (Figure 3, e), (Figure 3, middle, structure a), (Figure 3, structure b).

Line 242:  The authors write: ‘those which appear to robustly form a defined and 243 invariant secondary structure ‐ we would list essentially three regions’. You should list the names or numbers of the three regions at this point in the document.(SLI, the upper part of domain III, base of the domain III and the double pseudoknot).  It would make the rest of the paragraph easier to follow for the reader.

Line 332 ‘Of course, we must be aware that probably only the high Mg2+ 332 concentrations used in the above studies have allowed at all to show us the amazing IRES and ribosome structures we now know’. This not a good sentence and should be modified

Line 505: ‘On the contrary, binding of the HCV IRES to the 40S subunit and to eIF3 rather appears to be synergistically’ This is not a good sentence and should be modified.

‘Lines 328-525

.Thereby, the association of some molecules of the HCV IRES RNA with some 40S subunits can 529 occur within seconds, but saturating binding of most molecules of IRES RNA and 40S subunits in the 530 populations takes about 40 ‐ 80 s [138]. The presence of miR‐122 can greatly accelerate and enhance this process [57]. Formation of the first wave of complete ribosomes requires 4 ‐ 6 minutes [57, 532 144]. Between 6 and 10 minutes, the first wave of newly formed 80S complexes leaves the initiation site, and the second wave of complexes forms [57].

The comparison of temporal translation initiation process would be easier to interpret by the reader if the translation steps were described using the same terms. For example for translation the temporal stages are ‘Association with the 40S subunits’ and ‘saturating binding with the 40S’. and in the second part, with miR-122. The stages are ‘first wave of 80S’, and second wave of complexes.

Line 631 ‘In contrast, the NFAR protein complex (NF90, NF45 and RHA) that binds to the HCV 632 5´‐  and 3´UTRs [197] is for sure involved in HCV replication but obviously not in translation 633 regulation [198]. This is not a good sentence and should be revised.

Line 647: ‘a RNA chaperone function’ should be ‘an RNA’

Author Response

Reviewer 1

Comments and Suggestions for Authors

This review provides an in-depth description of how Hepatitis C virus translation is regulated by several regions of the HCV genome and the history of their identification. It also describes alternative structures formed by the 5’ UTR and IRES and how they might be influence by annealing of miR-122, and also provides novel predictions and interpretations of potential 5’ UTR RNA structures and new models of how miR-122 might modulate them. It then provides an overview of how the HCV IRES initiates translation under ‘normal’ conditions and how its regulation is modified during cellular stress or antiviral responses that down regulate translation. It also describes the canonical translation factors involved and how IRES trans‐acting factors (ITEFs) might regulate the HCV IRES and mediate interactions between the HCV genomic 5’ and 3’ untranslated regions. The review is extremely thorough and generally well written. I suggest only a few minor revisions to the content and English as listed below.

Minor issues:

Line 34: the person infected with ʺJapanese Fulminant Hepatitisʺ isolate 1 did not die. I believe they were treated and cured in spite of the high HCV replication levels displayed during that specific infection. This sentence should be modified based on the information in reference 3.

Reply:

Thank you very much, we have corrected the citation to Farci et al., 1996 (NEJM) and modified the sentence.

Line 44. ‘high error rate of the viral replicase can easily give rise to resistance mutations’. While this is true the low fitness of some resistant viruses has limited the impact of resistance thus far during patient treatment. Resistance might be an issue in the future, but it does not seem to be a huge issue now. It is very different than HIV treatment resistance. This sentence might be tempered to reflect this.

Reply:

Thank you very much, we have modified the sentences accordingly (with corresponding citations), pointing out the relatively low resistance of nucleoside inhibitors like Sofosbuvir due to the conserved active center of the replicase.

Line 55, ‘cap’ nucleotide should be capped nucleotide

Reply:

We guess Reviewer 1 refers to line 99 of the first manuscript version: "... the 5´end of the HCV genomic RNA has no cap nucleotide attached ...". Well, in our view the "capped" nucleotide is the first transcribed nucleotide which is post-transcriptionally processed and capped (by addition of N7-Methyl-Guanosin via a triphosphate bridge). In contrast, the "cap" nucleotide actually refers to the added N7-Methyl-Guanosin itself, which is a characteristic of capped mRNAs. This term is also used in most papers cited in PubMed. Therefore, we would prefer to keep the term "cap" nucleotide in this context.

Line 190 ‘In addition, a negative hybridization between sequences in the core‐coding region and the 191 region between 5´UTR SLs I and II (red in Figure 1) had been described’ what is a NEGATIVE hybridization? Please clarify.

Reply:

OK, we absolutely agree. We have modified the sentence to: "In addition, a hybridization between sequences in the core-coding region and the region between 5´UTR SLs I and II (red in Figure 1) had been described, which has a negative effect on translation efficiency."

Line 224. Other reports predicted and analyzed the 5’ UTR structure and predicted miR-122’s influence on the IRES structure and function and should be referenced. PMID 30053137, PMID 30941417. In addition, the alternative structure vs the canonical structure of SLI and SLII has not been confirmed experimentally. In papers that analyzed the RNA structure using SHAPE the results were consistent with both the canonical and the alt structures.

Reply:

Thank you very much, we absolutely agree. We have now included both additional citations (also in the legend to Figure 3). The passage in the main text has now been modified and extended to reflect these thoughts:

"An alternative predicted 5´UTR structure shows a refolded, alternative SL II (SL IIalt) (Figure 2B), which is as consistent with the reported experimental structure mapping results as is the canonical structure {Brown, 1992;Pang, 2011;Schult, 2018;Chahal, 2019}. In this alternative fold of the 5´UTR sequences between SL I and SL III (Figure 2B), the two miR-122 binding sites are largely hidden within double-stranded RNA structures and therefore are less accessible compared with the canonical structure (Figure 2A). In the presence of miR-122, the alternative SL IIalt reforms to adopt the canonical structure, allowing efficient translation and stabilization of the genome {Amador-Canizares, 2018;Schult, 2018;Chahal, 2019}. Interestingly, also mutations that render HCV replication miR-122 independent favor the canonical structure {Amador-Canizares, 2018;Schult, 2018}. "

The figure reference should be a consistent format. In places the sections of Figure 3 are referenced with (Figure 3, e), (Figure 3, middle, structure a), (Figure 3, structure b).

Reply:

OK, we have modified the Figure reference format to capital letters, and we have tried to standardize the referencing from the main text.

Line 242:  The authors write: ‘those which appear to robustly form a defined and 243 invariant secondary structure ‐ we would list essentially three regions’. You should list the names or numbers of the three regions at this point in the document.(SLI, the upper part of domain III, base of the domain III and the double pseudoknot).  It would make the rest of the paragraph easier to follow for the reader.

Reply:

OK, thank you very much! We have now included a short summary like suggested: "In brief, these are the SLI, the upper part of domain III, and the base of the domain III with the double pseudoknot."

Line 332 ‘Of course, we must be aware that probably only the high Mg2+ 332 concentrations used in the above studies have allowed at all to show us the amazing IRES and ribosome structures we now know’. This not a good sentence and should be modified

Reply:

OK, thank you very much! We have now modified this very enthusiastic sentence to a more moderate version: "The high Mg2+ concentrations used in the above studies have allowed to demonstrate the IRES and ribosome structures previously reported (see above)."

Line 505: ‘On the contrary, binding of the HCV IRES to the 40S subunit and to eIF3 rather appears to be synergistically’ This is not a good sentence and should be modified.

Reply:

OK, thank you very much! After checking back the data in the Pestova 1998 and Kieft 2001 papers, we now only cite Kieft et al. (2001) and state: "Surprisingly, binding of the HCV IRES to the preformed 40S-eIF3 complex is essentially not impaired by the presence of eIF3 on the 40S ribosomes, but the presence of eIF3 appears to facilitate IRES binding to the 40S subunit {Kieft, 2001}.".

‘Lines 328-525

.Thereby, the association of some molecules of the HCV IRES RNA with some 40S subunits can 529 occur within seconds, but saturating binding of most molecules of IRES RNA and 40S subunits in the 530 populations takes about 40 ‐ 80 s [138]. The presence of miR‐122 can greatly accelerate and enhance this process [57]. Formation of the first wave of complete ribosomes requires 4 ‐ 6 minutes [57, 532 144]. Between 6 and 10 minutes, the first wave of newly formed 80S complexes leaves the initiation site, and the second wave of complexes forms [57].

 The comparison of temporal translation initiation process would be easier to interpret by the reader if the translation steps were described using the same terms. For example for translation the temporal stages are ‘Association with the 40S subunits’ and ‘saturating binding with the 40S’. and in the second part, with miR-122. The stages are ‘first wave of 80S’, and second wave of complexes.

Reply:

OK, thank you very much for this comment! We have been reading through the entire paragraph and now have changes some phrases according to the suggestions of the Reviewer. We hope that these changes make our thoughts more clear.

Line 631 ‘In contrast, the NFAR protein complex (NF90, NF45 and RHA) that binds to the HCV 632 5´‐  and 3´UTRs [197] is for sure involved in HCV replication but obviously not in translation 633 regulation [198]. This is not a good sentence and should be revised.

Reply:

OK, we have revised this sentence to:  "In contrast, the NFAR protein complex (NF90, NF45 and RHA) appears not to be involved in HCV translation regulation {Li, 2014}, even though it binds to both the HCV 5´- and 3´UTRs and is involved in replication {Isken, 2007}."

Line 647: ‘a RNA chaperone function’ should be ‘an RNA’

OK, done.

Thank you very much for your  helpful comments and suggestions for improvements! 

Reviewer 2 Report

  1. HCV IRES consists of 5’UTR, core and 3’UTR (see. Korf M, et al., Inhibition of hepatitis C virus translation and subgenomic replication by siRNAs directed against highly conserved HCV sequence and cellular HCV cofactors. J Hepatol 2005;43:225-234.; Kanda T, et al. Small interfering RNA targeted to hepatitis C virus 5' nontranslated region exerts potent antiviral effect. J Virol. 2007 Jan;81(2):669-76.). Like this, authors already made figure 1. Authors should revise their abstract section.
  2. In Introduction section, HCV does not cause severe hepatitis. (Kanda T, et al. Acute hepatitis C virus infection, 1986-2001: a rare cause of fulminant hepatitis in Chiba, Japan. Hepatogastroenterology. 2004 Mar-Apr;51(56):556-8.) Authors should extensively revise their manuscript.
  3. In Figure 1 legend, authors should add the reference for ARF. See. Basu A, et al. Functional properties of a 16 kDa protein translated from an alternative open reading frame of the core-encoding genomic region of hepatitis C virus. J Gen Virol. 2004 Aug;85(Pt 8):2299-2306.

Author Response

Reviewer 2

Comments and Suggestions for Authors

HCV IRES consists of 5’UTR, core and 3’UTR (see. Korf M, et al., Inhibition of hepatitis C virus translation and subgenomic replication by siRNAs directed against highly conserved HCV sequence and cellular HCV cofactors. J Hepatol 2005;43:225-234.; Kanda T, et al. Small interfering RNA targeted to hepatitis C virus 5' nontranslated region exerts potent antiviral effect. J Virol. 2007 Jan;81(2):669-76.). Like this, authors already made figure 1. Authors should revise their abstract section.

Reply:

Thank you very much for this comment! Accordingly, we have now revised the abstract and included in the first sentence: "Translation of the Hepatitis C Virus (HCV) RNA genome is regulated by the internal ribosome entry site (IRES), located in the 5´-untranslated region (5´UTR) and part of the core protein coding sequence, and by the 3´UTR".

In Introduction section, HCV does not cause severe hepatitis. (Kanda T, et al. Acute hepatitis C virus infection, 1986-2001: a rare cause of fulminant hepatitis in Chiba, Japan. Hepatogastroenterology. 2004 Mar-Apr;51(56):556-8.) Authors should extensively revise their manuscript.

Reply:

Thank you very much for this very helpful comment. We have now corrected this sentence, stating that "Acute infection can result in severe liver damage, in rare cases even resulting in death (Farci, 1996;Kanda, 2004)".

In Figure 1 legend, authors should add the reference for ARF. See. Basu A, et al. Functional properties of a 16 kDa protein translated from an alternative open reading frame of the core-encoding genomic region of hepatitis C virus. J Gen Virol. 2004 Aug;85(Pt 8):2299-2306.

Reply:

Ok, done. We have included the first four citations on ARF in the legend of Figure 1, including Basu et al., 2004. 

Round 2

Reviewer 2 Report

All queries were fixed.